# Forecasting Covid-19 Dynamics in Brazil: A Data Driven Approach

**DOI:** 10.3390/ijerph17145115

**Published:** 2020-07-15

**Authors:** Igor Gadelha Pereira, Joris Michel Guerin, Andouglas Gonçalves Silva Júnior, Gabriel Santos Garcia, Prisco Piscitelli, Alessandro Miani, Cosimo Distante, Luiz Marcos Garcia Gonçalves

**Affiliations:** 1Department of Computer Engineering and Automation, Federal University of Rio Grande do Norte, Natal 59078-970, RN, Brazil; igorgad@gmail.com (I.G.P.); jorisguerin.research@gmail.com (J.M.G.); andouglas.silva@ifrn.edu.br (A.G.S.J.); 2Department of Computer Science, Federal Institute of Rio Grande do Norte, Mossoro 59628-330, RN, Brazil; 3Institute of Biological Sciences, University of Brasilia, Distrito Federal 70910-900, Brazil; gasagarcia@gmail.com; 4Euro Mediterranean Scientific Biomedical Institute (ISBEM), 1040 Bruxelles, Belgium; priscofreedom@hotmail.com; 5Department of Environmental Sciences and Policy, University of Milan, 20133 Milan, Italy; alessandro.miani@gmail.com; 6Institute of Applied Sciences and Intelligent Systems, 73100 Lecce, Italy; cosimo.distante@cnr.it

**Keywords:** time series prediction, Covid-19 pandemic, modified auto-encoder, data-driven

## Abstract

The contribution of this paper is twofold. First, a new data driven approach for predicting the Covid-19 pandemic dynamics is introduced. The second contribution consists in reporting and discussing the results that were obtained with this approach for the Brazilian states, with predictions starting as of 4 May 2020. As a preliminary study, we first used an Long Short Term Memory for Data Training-SAE (LSTM-SAE) network model. Although this first approach led to somewhat disappointing results, it served as a good baseline for testing other ANN types. Subsequently, in order to identify relevant countries and regions to be used for training ANN models, we conduct a clustering of the world’s regions where the pandemic is at an advanced stage. This clustering is based on manually engineered features representing a country’s response to the early spread of the pandemic, and the different clusters obtained are used to select the relevant countries for training the models. The final models retained are Modified Auto-Encoder networks, that are trained on these clusters and learn to predict future data for Brazilian states. These predictions are used to estimate important statistics about the disease, such as peaks and number of confirmed cases. Finally, curve fitting is carried out to find the distribution that best fits the outputs of the MAE, and to refine the estimates of the peaks of the pandemic. Predicted numbers reach a total of more than one million infected Brazilians, distributed among the different states, with São Paulo leading with about 150 thousand confirmed cases predicted. The results indicate that the pandemic is still growing in Brazil, with most states peaks of infection estimated in the second half of May 2020. The estimated end of the pandemics (97% of cases reaching an outcome) spread between June and the end of August 2020, depending on the states.

## 1. Introduction

The world population is being rapidly infected by the SARS-COV-2 virus pandemic, also known as Covid-19 [1]. This virus has spread very fast throughout the countries, with a very high contagion rate, reaching all continents in just over three months since the first confirmed case in Asia [2]. The numbers have grown exponentially, reaching six-million of confirmed cases and almost half a million deaths. Distancing and social isolation rules have been used as the only alternative in order to contain the progress of the disease. The flattening of the virus spreading curve, which can be modeled using different approaches [3,4,5,6,7], is the main goal that is strictly related to the rules mentioned. If this does not happen, the number of deaths would skyrocket, as it was recently experimented in countries, such as the USA, Brazil, Italy, Spain, France, and the UK. Several warnings about this have been spread in the literature, for example, in the beginning of March 2020, Fanelli [3] explained that: “In Italy and in other countries that will be facing the epidemic surge soon, this is quite possibly only achievable through a cooperative and disciplined effort of the population as a whole”. Successful example of curve flattening have already been seen in Portugal, Germany and South Korea, among others. Other countries, such as New Zealand, managed to limit the spread by completely closing their borders and imposing a complete lock-down to their people. In these countries, the counter-measures provide a certain breath to governments in order to allow the health systems of each country to meet local needs, or to be able to wait for other solutions, such as the development of vaccines (not yet existing so far for the SARS-CoV-2 virus). The medical solutions to minimize or stop the pandemic include in-silico analysis of the SARS-CoV-2 genome [8], aiming to study its weaknesses in an attempt to better understanding it and develop treatments and vaccines.

In parallel, studies aiming to determine the virus dynamics, its geographical distribution, and the peaks of the pandemic in given regions are necessary. Particularly, our study is justified for the correct planning of immediate actions to be taken by states management, such as estimating the required number of hospital beds or changing the counter measures strategy for smoothing the pandemic effects. Several attempts to model the spread of the virus have been recently conducted [5,6,7,9,10,11,12,13], including machine learning approaches [14]. However, we found that many of these methods rely on parameters that are dependent on the advance of the spreading and on the regional context [5,6,7,14]. In order to avoid such dependence on uncertainty parameters, this paper proposes a solution to the problem of virus dynamics prediction, which depends only on past and current data. This approach, as it only requires contamination data, can be straightforwardly applied to new regions without further parameter estimation. For this reason it is particularly relevant to forecast global pandemics such as Covid-19, which is affecting almost every region of the globe in a different way.

Regarding the evolution of Covid-19, many time series data are partially available, and in light of the recent successes of neural networks in time series prediction, using artificial intelligence seems a good strategy to predict future data. We argue that a data driven technique can be used to infer the pandemic dynamics from raw data, including future events such as the date of the peak, number of cases and deaths, and the end of the pandemic. Hence, this work tests and validates the possibility of using deep learning tools, to create models of dissemination and to predict the pandemics events in different regions, which can be used alternatively or in addition to traditional models [5,6,7]. We discuss our results in comparison to those obtained with traditional methods and found that they work at least with the same precision in predicting the pandemics events. This paper follows a preliminary study that was carried out in Italy, based on China data with satisfactory results, but with space for improvement [9,15]. An approach based on Long Short Term Memory for Data Training (LSTM) has been initially tested by our team in Brazil, and demonstrated problems that are discussed in Section 2.2.

The main contribution of this work is to present a way to train a Modified Auto-Encoder (MAE) to forecast virus spreading. The MAE demonstrated better results than the preliminary LSTM approach and was thus chosen as our final model for predicting data from Brazilian states. The obtained forecasts are reported at www.natalnet.br/covid, which is updated frequently. Different countries and states reacted to the pandemic differently, taking different actions as explained earlier. For this reason, there is no guarantee that using data from any given country for training could generalize well to any state for prediction. In order to overcome this problem, we propose an initial clustering of the different countries’ data, based on Early Mortality, Days until 10×, and Early Acceleration features. Subsequently, different prediction networks are trained within each cluster, using countries that have a more advanced stage of the pandemic than Brazil, e.g., China, Italy, Spain, and the United States. Each networks is then used to make predictions on the Brazilian states that belong to the cluster on which it was trained. We have provided comparison with traditional approaches as SIR and SEIR that will be presented at results and discussion Sections. The results are promising, besides the pandemics is still in a growing situation in Brazil.

Based on the results reported in this paper, we could verify the applicability of data driven methods to model the Covid-19 dynamics. With this approach, which deals with regional aspects of the pandemic, city managers can get more precise information to help them in planning their actions. Complementary data about peak prediction and estimated numbers have shown the applicability of our approach to Brazilian states with success. We underline that the findings reported in this paper come from estimated data and cannot be completely guaranteed as being the final truth. However, they are important because they allow managers and even population, to have an idea about what the future holds for the pandemic dynamics. We hope that, using the forecastings of the pandemic curves presented in this paper, better decisions can be taken to help protect the population. Additionally, we underline that, for a better real-time assessment of the pandemic dynamics, these data driven models should be constantly (at least weekly) updated with new data.

## 2. Materials and Methods

This work aims to develop a method for predicting the dynamics of transmission of viral epidemics, using artificial intelligence methods on contamination data from more advanced regions. Deep Learning techniques are studied and implemented, aiming to learn the dynamics of the pandemics using data from other locations. This approach is then applied to the specific case of Brazil. We start by describing traditional approaches to set a baseline for comparison, and then detail the different components of the data driven method retained.

### 2.1. Modeling Virus Dynamics (Traditional Approaches)

The spread and contamination of the Covid-19 virus is not entirely random and it follows certain patterns. These dynamics can vary across different regions as they depend on parameters, such as pollution, demographic density, average age of the population, among others. Analyzing the actions taken to fight the virus, in both the social and economic spheres, there is a need for more realistic epidemiological data. Indeed, the use of local models, taking into account the reality of each region, state or municipality, can allow the authorities to take coherent decisions. Therefore, it is assumed that the spread of the virus follows some statistical model, which parameters can be tuned to represent different situations.

Approaches to model the behavior of infectious diseases, such as SEIR, have been used to model the epidemic of COVID-19 [5,16]. In these approaches, the phase transitions of the disease are modeled as instantaneous rates in differential equations or as probabilities of transition in discrete time differences or matrix equations. These models provide accurate estimates of the position of the equilibrium points, when the rate at which individuals enter each stage is equal to the rate at which they exit. However, they do not accurately capture the distribution of the time an individual spends at each stage; therefore, they do not accurately capture the transitory dynamics of epidemics. Actually, the SEIR model has been tested for Italy [6] to model the dynamics of the COVID-19 epidemic. It has been shown to underestimate peak infection rates (by a factor of three using published parameter estimates based on the progress of the epidemic in Wuhan) and to substantially overestimate the persistence of the epidemic after the peak has passed [5].

Other approaches, such as SIR [12,13,17], SEIRD [6], and SEITR [18], are also helpful to understanding the Covid-19 dynamics. Nonetheless, the lack of ground truth data prevents us from determining which of these models is the most precise. Despite somehow representing the Covid-19 dynamics, some of these traditional models (SIR, SEIR, SEITR, SEIRD) must be improved so that they can be applied with higher precision to the study of the new virus, as they have been shown to present some issues on the recent works cited above. In this work, besides discussing the main advances of the contributions in this direction, these traditional models are compared to ours, which is a purely data-driven approach. Some preliminary studies on the above methods have been conducted for better understanding of the Covid-19 dynamics. Adaptations of the SIR model have also been used in Brazil, including parameters that comprise the effects of social distancing measures [19]. In these approaches two variations of the SIR model are proposed considering conditions as for example the current social distancing policy imposed by the government for an indefinite time, which is somewhat a similar rule to our approach. Nonetheless, our approach do not need to explicit know the condition.

In fact, we verified that Covid-19 is a virus that cannot be modeled perfectly with any specific traditional model because of the influence of several factors on its dissemination speed. Mainly, it is difficult to model its behavior because of the non-linearity of infection data caused by under-notifications and also the lack of effective and constant counter measures, which changes all the time as the infection spreads. For these reasons, it seems appealing to apply AI-based methods. As a first test, we start by implementing an LSTM, one of the default neural network models for analyzing time series data, in the next section.

### 2.2. Long Short Term Memory for Data Training (LSTM)

Several neural network models can be used to solve problems of time series estimation. Recurrent neural networks (RNN) are a family of architectures that contain recurring feedback connections, which define an internal state, or short-term memory. This memory makes them suitable for modeling sequential or time series data [20]. To this end, a standard RNN keeps a vector of activation parameters at each time step, especially when short-term dependencies are included in the input data. However, when trained with gradient descent algorithms, learning the long-term dependencies that are encoded in data becomes difficult due to the vanishing gradient problem. This is solved while using a specialized neuron for long-term memory that keeps a constant reverse flow in the error signal, allowing it to learn long-term dependencies. This approach was presented by Hochreiter [21] and it is known as LSTM (Long Short Term Memory).

In this way, a LSTM network is a kind of RNN architecture, having a recursive branch for modeling time series and solving the vanishing gradient problem. To do so, it uses a memory cell that is able to represent long-term dependencies in the time series, composed of four neural units: input, output, forgetting and the self-recurring neuron (Figure 1a). These units are responsible for controlling the interactions between different memory units. Specifically, the input unit controls whether the input data can modify the state of the memory cell or not. On the other hand, the output unit controls whether or not it can change the state of other memory cells.

Mathematically, considering the output gates ($f_{t}$, $i_{t}$, $o_{t}$ and $\tau_{t}$) shown in Figure 1a, we have:(1)$$f_{t}=\sigma \left(X_{t}U^{f}+S^{t-1}W^{f}+b_{f}\right)$$
(2)$$i_{t}=\sigma \left(X_{t}U^{i}+S^{t-1}W^{i}+b_{i}\right)$$
(3)$$O_{t}=\sigma \left(X_{t}U^{o}+S^{t-1}W^{o}+b_{o}\right)$$
(4)$$\tau_{t}=tanh\left(X_{t}U^{c}+S(t-1\right)W^{c}+b_{c})$$
(5)$$C_{t}=C_{t-1}⊗f_{t}⊕i_{t}⊗C_{t}^{\prime }$$
(6)$$S_{t}=O-t⊗tanh\left(c_{t}\right)$$
where, **U**, **W**, and **b** are respectively the input weights, recurrent weights, and biases; **X** is the input; **S** is the hidden output; **C** is the cell state; and **t** is the time step. The σ and τ are the activation functions of the output gates. In the classical LSTM model, the first one is the Sigmoid function and the second one is the hyperbolic tangent function. There are several types of activation functions that could be used in LSTM architectures [22].

According to Sagheer [20], despite the advantages of the LSTM architecture, its performance for time series problems is not always satisfactory. The shallow LSTM architecture may not represent the complex features of sequential data efficiently, especially if they are used to learn data from long-range time series with high non-linearity, which is the case for Covid-19 data. Other RNN architectures based on LSTM have been created in order to overcome this problem. We tested two approaches proposed by Sagheer: DLSTM [23] and LSTM-SAE [20], while using Covid-19 data from China provinces (daily number of cases and cumulative number of cases).

The LSTM-SAE and DLSTM blocks are shown in Figure 1b,c respectively. Basically, both blocks are composed of stacked LSTM layers, which increase the depth of the network. Besides that, the LSTM-SAE configuration uses an auto-encoder to initialize the weights of each LSTM layer. In our application, we used only one hidden layer for this setup, but it is possible to use more layers and more auto-encoders, as shown on the original paper. We trained three models, one LSTM, one DLSTM, and one LSTM-SAE, in order to select the best architecture for the Covid-19 problem. These models were trained using data from all China provinces except Hubei (that was used for testing). We evaluated which model generalized best to the dataset available using the MAPE metric. Finally, we used a dynamic prediction, where the model is updated for each new predicted value. This method improves the forecast due to the incorporation of data from other countries or regions. The training parameters and results metrics are shown in Table 1.

Figure 2a,b show the results for the three trained models for Hubei (province of China). As shown in Table 1, the best model was LSTM-SAE, thus being chosen as the model to forecast other regions or countries.

On the one hand, despite the devastating effects of the pandemic, three months of data is a relatively short period of time for training complex time series prediction models without overfitting, which has been reported as one of the main problems for training LSTMs (see Section 4). On the other hand, this pandemic is the first large scale global pandemic that our generation has to face and there are not yet standardized guidelines for countries on how to react to such an event. For this reason, responses to the pandemic have varied widely throughout the different regions and countries worldwide, thus creating a huge variability in the available data. Hence, we propose to conduct a preliminary study that consists in grouping countries and regions with similar early responses. In this way, smaller specialized networks can be trained for each cluster, and we hope that, by learning on more consistent data, our models could generalize better without overfitting to the training data. Additionally, we found a better model (MAE) that is used, instead of LSTM, on data that result from the clustering approach that is described next.

### 2.3. Preliminary Clustering: Brazilian States in the Global Context

The objective of this paper is to train a predictive model for Brazil, as well as some distinct models for each of the groups of Brazilian states. Hence, the proposed clustering pipeline considers both entire countries and smaller regions as entries. The input data used in this preliminary study are all countries available in the JHU dataset [24], Chinese and Canadian provinces, American, Australian, and Brazilian states [25] as well as Italian Regions [26].

The approach used for identifying that countries present similar early responses to Covid-19 is inspired by the literature in this area [27]. First, we define the *outbreak date* of a country to be the day at which it registered five confirmed cases per million inhabitants. Normalizing by the population of the region helps to characterize the true response of a country, avoiding to give more weight to highly populated countries. Figure 3 shows the number of accumulated deaths per million inhabitants for the different Brazilian states on the first of May 2020.

We start with the preprocessing scheme to be applied on this dataset. A seven-day arithmetic moving average is first calculated to each time series of the dataset. This is done to deal with the seasonality that is observed in data, i.e., higher variability during the weekends. After filtering, a feature representation containing three characteristics is computed for each time series. These features are:Early Mortality: weekly number of deaths 14 days after the outbreak, divided by the number of confirmed cases, in the week of the outbreak. A two weeks period was used because it is the time required to know the outcome of a contamination.Days until 10x: the number of days it takes to multiply the confirmed cases by 10, from the day of the outbreak.Early Acceleration: if we denote $\Delta_{W_{0}W_{1}}$ as the percentage increase of confirmed cases from the week of the outbreak to the week after, and $\Delta_{W_{1}W_{2}}$ as he percentage increase from the 1st to the 2nd week after the outbreak, then the early acceleration is defined by:
(7)$$earlyAccel=\Delta_{W_{1}W_{2}}/\Delta_{W_{0}W_{1}}.$$

The values of these features for the different Brazilian states are shown in Figure 4.

Subsequently, the clustering pipeline is applied to the former feature representation to group the different countries/regions together. To do that, a Uniform Manifold Approximation (UMAP) embedding [28] is applied to generate a two-dimensional clustering friendly feature space. UMAP is an unsupervised embedding method that tends to preserve the global distances present in the initial dataset. This lower dimensional feature space not only facilitates the visualization and interpretation of data but also tends to improve clustering results for algorithms where the number of clusters is unspecified. In practice, UMAP is used with n_neighbors=15 and min_dist=0. However, UMAP only produces a new embedded space and does not generates directly the clusters assignments, which are needed to select the countries for training our neural network models.

To solve this issue, we use the scikit-learn [29] implementation of Affinity Propagation [30] with a damping factor of 0.8, applied to the UMAP embedded space. The results from this preliminary clustering procedure are further presented in Section 3.3.

Therefore, our clustered data series is ready for the MAE training and prediction procedures, depending on the phase. In practice, to forecast contamination data of a given Brazilian state, we use the time series data of the countries/regions that belong to the same cluster, and that are at a more advanced stage of the pandemic. In this section, the clustering approach adopted to characterize the responses of the different countries is explained and the details of the training process are explained in the following section.

### 2.4. Modeling Time-Series with Modified Auto-Encoders

We propose using a set of Modified Auto-Encoders (MAE) to forecast time-series data regarding the number of daily confirmed cases of Covid-19 un order to model the transmission dynamics of the SARS-COV2 virus in Brazil. An auto-encoder is a specific neural network architecture that is trained to copy its input to its output [22]. In this way, the auto-encoder generates a hidden representation that describes useful properties of the input data.

The network architecture can be divided in two parts: an encoder function h=f(x), which maps the input data *x* to the hidden representation *h*, and a decoder function $\hat{x}=g\left(h\right)$ that attempts to approximate the input $\hat{x}$ from the hidden representation. With the use of the stochastic gradient descent strategy to train neural network architectures, the auto-encoder mapping functions can be generalized to stochastic mappings, such as $p_{encoder}\left(h|x\right)$ and $p_{decoder}\left(\hat{x}|h\right)$.

The hidden representation, also called latent space, generated by the mapping $p_{encoder}\left(h|x\right)$ contains a stochastic representation of the probability distribution of the input data and can be used for dimensionality reduction [22], feature learning [22], and also in generative models when combined with latent variable models [31].

#### 2.4.1. The Modified Auto-Encoder proposal

Auto-encoders can also learn useful properties from time-series if a sequence is applied to its inputs. Such properties may be used to forecast the next samples of the given input sequence. In this way, we propose to modify the traditional auto-encoder architecture in order to employ an extra output derived from the latent space. Therefore, while the traditional output of the auto-encoder is trained to approximate the input values, the extra output is trained to approximate the next sample of the sequence given to the input of the auto-encoder.

Consider *X* a sequence, such as $X=x_{1},x_{2},\ldots ,x_{n}$, the latent space vector *H* is obtained with the mapping $p_{encoder}\left(H|X\right)$ and the traditional output of the auto-encoder is obtained with the mapping $p_{decoder}\left(\hat{X}|H\right)$. The extra output added to the auto-encoder model tries to approximate $x_{n+1}$ with the mapping $p_{predictor}\left(x_{n+1}|H\right)$.

In order to increase the latent space dimension without increasing the input sequence, we apply three auto-encoders in parallel and aggregate their latent space before computing the predictor output. Such Modified Auto-Encoder (MAE) architecture is depicted in Figure 5.

The predictor output, the input-samples and the decoder output have one, eight, and eight units, respectively. Each encoder, latent space and decoder has 32, four, and 32 units, respectively. The output of each decoder is averaged to create the total decoder output. The latent spaces of each auto-encoder are concatenated prior the final computation of the predictor output. We train the modified architecture with the mean squared error loss function and the Adam optimizer.

#### 2.4.2. Data Processing

Let us consider the epidemic curve a time-series that models the advance of an epidemic by measuring the number of new confirmed cases of Covid-19 on a daily basis. Hence, we first apply a moving average filter of size 3 to deal with the variability of data related to the amount of tests available and delays in reporting between other problems.

We compute the input examples by dividing the whole epidemic curve in overlapped segments of 8 days, shifted one day from each other, with the 9th day being the value to be forecast by the MAE. Each example is normalized by dividing its values by its maximum value. A set of 10 examples is taken from the most advanced places in each cluster in order to form a batch of examples.

We compute the difference between the number of cases sampled at the day of peak occurrences and the last number of cases reported in order to evaluate the most advanced places in the epidemic timeline. If this difference is positive, the number of daily cases started to decrease, meaning that such place passed the peak number of cases and is more advanced in the epidemic timeline.

#### 2.4.3. Forecasting New Daily Cases

We start by applying the same moving average filter of size 7 to the epidemic curve of the Brazilian states as depicted in details in Section 2.3 and we then perform the forecasting on these clustered time series data in two phases in order to forecast the Brazilian epidemic curve.

The first phase uses existing data to feed the network, and the forecast value is one-step ahead of the current example. In the second phase, referred to as multi-step ahead, we use the predicted value of the *i*-th step to forecast the value of the (i+1)-th step. In this way, it is not necessary to have existing data for the second phase of forecasting, allowing for us to forecast the epidemic behaviour several days ahead and identify the probable date of the peak number of daily cases, which might indicate a drop in the number of occurrences. Notice that this peak or the end of the pandemic might be subject to some displacements due to problems in the data, so a final step needs to be applied in order to verify the peaks for all states. This is done by fitting a distribution curve on the output data, as described next.

### 2.5. Final Approximation for the Covid-19 Curves

Despite that we discuss below the hardness of finding a curve that mathematically and exactly represents the Covid-19 dissemination, the main and most important reason for trying an approximation to this curve is that it allows for defining useful information, such as verifying the peak and estimating the end of the pandemics. Moreover, it can generate more realistic number of cases to some degree of precision, thus being of importance. To determine the end of the pandemic or the peak are two of these advantages, as it is supposed that epidemics obey certain statistical rules [7], to some degree of precision. In this work, we verify the peaks after approximating the final predicted curve while using some statistical procedure.

In relation to modeling Covid-19 using statistical distributions, it has been discussed that this is a somewhat difficult task. Actually, the Covid-19 curve can not be considered a Gaussian probability distribution [32]. In fact, it is argued that the shape of a normal distribution is a histogram that is a transformation of probability density against values of a single variable, while the Covid-19 contagion curve is a transformation of the values of one variable (confirmed cases) according to a second variable (time). Accordingly, the curve is not a distributions in the sense usually meant in probability and statistics. Nonetheless, one can visually notice that the curve of daily confirmed cases × time (day) looks like a distorted Gaussian, and besides hard of being exactly determined it can actually be approximated by some distributions, such as the normal (rarely), pearson, logistic, logNormal, burr, and gamma, among others. For the sake of confirming or ratifying the estimated peak and pandemic end, we thus conduct a final statistical procedure to the time series data output by the MAE models.

## 3. Results

In this section, we present the results of the experiments that were carried out to validate the proposed methods. We start by describing results from the literature for traditional approaches as well as the LSTM results obtained. These results are used as a baseline for comparison against our proposed approach. Then, results of the clustering procedure are shown in order to validate the approach used. Finally, we present the results obtained with the Modified Auto-Encoder model to forecast the epidemic curves of Covid-19 in Brazil, as well as the fitted distribution curves confirming the peaks obtained for all Brazilian states. The numbers obtained for Brazil, which are of direct interest to the population, are presented in for some states, and complete results can be found at www.natalnet.br/covid. We underline once more that these numbers are predictions, and that they might differ from the real values, as the pandemic dynamics evolves depending on how the population complies to social distancing rules and other counter measures. However, we will show that, to this day, the numbers predicted by our method are better than traditional methods.

### 3.1. SIR, SEIR, and SIRASD Results

The results using SIR and SEIR can be found in several applications running on the Internet [33]. Before discussing these results, we note that to this day, we could not find accurate results on Covid-19 long-term dynamics prediction while using these methods. However, notice that any long-term series approximation method is intended to be useful at pandemic time, as long-term forecasts may help managers to discuss different types and duration of confinement policies [19]. For example, the preliminary results reported by Bastos and Cajueiro in 1 April using SIR and SIAS are far from being accurate, as noticed in the first version of their work [19]. Indeed, they suggest that 30 million people will get infected in Brazil on 11 May (pandemic predicted peak) in the less infected people situation. Later, in their revised work evolving to SIRD and SIRASD approaches [19], the results are better; however, they are not precise yet. We will discuss our results in comparison with those results further.

Regarding the Northeast region in Brazil (our main interest region), the government uses a website maintained by UFRN for monitoring and predicting Covid-19 [33]. These forecasting are based on a modified SEIR model accounting for social distancing rules [34]. According to them, the epidemic started on 1 March and the symptomatic cases are predicted to end on 1 July. The peak of symptomatic people is predicted for 17 May, with 20 million contaminated people. By taking a closer look to the predictions, we can see that the Brazilian state RN was predicted to have 2039 confirmed cases on 30 April 2020, following the current scenario of social distancing, as shown in Figure 6 (printed from the website http://astro.dfte.ufrn.br/html/Cliente/COVID19.php). The true value for the confirmed cases on that day was actually of 1297, which is not so bad (less than 50% error). However, a greater disparity between values was acknowledged for the whole Brazil (Figure 7). Indeed, their prediction was about 742 thousand confirmed cases by the same day, whereas the actual number was 87,187 confirmed cases (according to the Health Ministry of Brazil—MS). We underline again that these predictions, although a bit exaggerated, are useful for the authorities to take decisions. They often represent the worst case scenarios of the pandemic. As it will be shown, our results tend to be more humble than the ones that were reported on this site and we sincerely hope that our predictions are still exaggerated.

### 3.2. Preliminary Results with LSTM

In our initial studies towards data driven approaches, we tested the possibility of using the LSTM-type RNN for determining Covid-19 dynamics for our region of most interest (Brazilian state RN), but it did not work as expected due to several factors. Mainly the under-notification of data made available by the governments of Brazil and its Federate States. Hence, more work is necessary for improving and testing this model, and adjusting it to predict the dynamics of the pandemic, including its various parameters. The problems with this architecture applied to COVID-19 data will be further discussed in the Section 4. Basically, the main drawback of the model is its inability to reset at certain time and bringing the values to zero (or close). Anyway, we used the LSTM-SAE to forecast three different places, with different phases of the disease: Italy (Figure 8), where the contamination curve is starting to decrease; Brazil (Figure 9), which is approaching the peak; and, the state of Rio Grande do Norte (Figure 10) that is about to reach the peak.

Although not responding perfectly, we notice some LSTM important features that can be seen on the charts. One is that the LSTM-SAE model could stabilize over time. By analyzing the daily results, the other LSTM models cannot return to zero and keep oscillating around some positive value. Because of this, when the value is accumulated it always increases. This issue is more apparent when the model is used for countries or regions that did not stabilize their cases. These limitations can be associated to the non-linearity of data, among other issues. Another point is that, as presented in previous work [20], the LSTM-SAE addresses the input data randomization in the LSTM block. The encoder-decoder model trained first feeds the hidden layer with initialization weights. It is possible that, because of that, the LSTM-SAE architecture presents the best results. More complete results using LSTM can be found at www.natalnet.br/covid.

### 3.3. Checking the Clustering Results (Input to MAE)

Before showing MAE results, this subsection presents and discusses the results from the preliminary clustering necessary for the better performance of MAE, which was presented in Section 2.3. To qualitatively evaluate the clusters obtained, lets use the two-dimensional (2D) UMAP embedding that is shown in Figure 11. Seven clusters were formed, and overall, they seem rather compact and distinct from each others. Although there is a slight overlapping between some pairs of clusters, this plot suggests that there was actually well defined groups within the different countries/regions, probably reflecting the types of actions taken by the governments to react to the early signs of the pandemic. Notice that we suppose and believe that countries from the same clusters should follow similar contamination curves.

A map of Brazil representing the clusters is shown at Figure 12 in order to visualize this preliminary classification and get some insight for Brazilian states. In that map we separate in the same color the states and countries that presented a similar reaction to the outbreak of Covid-19. The countries that are represented by hatches in the maps were either not sufficiently advanced at the time of the study or their time series produced numerical instabilities during feature computation. The states from USA, Australia, Italia, China, and Canada appear in the different clusters but are not represented in the maps to improve readability of the paper. The full results of the cluster assignment used in the training process can be found at www.natalnet.br/covid.

Finally, the values of the features of the different groups are presented in the form of violin plot on Figure 13. We can see, for example, that cluster 0 gathers the countries with higher *Days until 10x*, meaning that these countries/regions managed to contain the contagion early. In turn, Brazil belongs to cluster 1, which contains countries with high early acceleration and above average mortality rate.

We conclude this section by underlining the fact that the colors representing the clusters used for Figure 11, Figure 12 and Figure 13 are matching, meaning that a country in yellow on the map belongs to the yellow cluster on the UMAP plot and its statistics can be seen in yellow on the violin plot. In addition, we believe that all major centers of Covid-19 are represented in these maps, which provides sufficient material to train our models.

### 3.4. MAE Results

This Section presents the results that were obtained by applying the MAE architecture model to forecast the Covid-19 epidemic curves of Brazilian states of each cluster. Therefore, for each cluster, a MAE model was trained with the 10 most advanced countries of the cluster with the data available up to the day of this study, and the epidemic curves for the Brazilian states of the cluster were forecast. However, we note that the Brazilian states were only on clusters 0, 1, 2, and 3.

Here, we depict one state for each cluster. For an interactive visualization of all Brazilian states, you may refer to https://www.natalnet.br/covid.

In Figure 14, the daily and cumulative epidemic curves for the Sergipe state is displayed. The peak for the Sergipe state is predicted to happen on 21 May and it should reach little less than 8000 cases at the end of July.

Figure 15 depicts the epidemic curves for the São Paulo state. In this case, we predict the peak number of cases for May 28th and that the state would reach a total of 150,000 cases at the end of July.

Figure 16 depicts the epidemic curves for the Rio Grande do Norte state. The peak occurrence in daily cases is predicted to happen in 20 May and it should reach around 8500 total cases at the end of August.

In the last cluster, we derived the epidemic curves for the state of Santa Catarina (Figure 17). The peak occurrence is predicted to happen in May 21st and it is supposed to reach around 9000 cases at the end of August.

From the epidemic curves that are illustrated above, we verify that each state has its behavior associated to the cluster that it belongs. States from cluster 0 generally present a steep peak, but a very low number of daily cases, which indicates that the epidemic is starting and evolving fast, but will not present an elevated number of daily cases.

The cluster 1 presents a different behavior. Generally, states from cluster 1 presents a steep peak with an elevated number of daily cases, meaning that the transmission dynamics is happening much faster than cluster 0. In the meantime, the predictions show that states from cluster 1 are close to reach the peak number of occurrence of daily cases and should have its occurrence of daily cases decaying very fast.

The states from cluster 2 present a slower rate of transmission dynamics if compared to states from the cluster 1 and, according to the date expected for the peak number of daily cases, these states still did not reach the peak number of occurrences.

States from cluster 3 present the slowest transmission dynamics and tend to have their number of daily cases decaying slowly.

We also indicate, in Table 2, the date of the peak occurrence of cases, the date that it will reach 97% of the total number of cases, the total number of cases, and a peak occurrence date obtained by fitting a probability distribution to the predicted curves as well as the curve used in the probability distribution fitting process. Examples of these curves are shown in Figure 18 and Figure 19, for the states of Rio de Janeiro and São Paulo, respectively, ratifying the peaks that are shown in Table 2. The other states curves can be found at www.natalnet.br/covid.

## 4. Discussion

As aforementioned, the LSTM based approaches did not work well for modeling the Covid-19 dynamics. The LSTM-SAE was also tried and performed a little better. There are some explanations for the lower performance of LSTMs. The first issue is related to data non-linearity that is caused by under sampling (every other day and then every day for example) and under notifications (numbers under real values), which have also been problematic for other countries than Brazil. Several countries are under-testing their population, making the number of reported cases below reality. Even for the countries that are doing massive testing, there are often delays between the real occurrences and notification. Another potential source of error is the randomization of the weights, which can be solved with LSTM-SAE [20]; however, the first issue still remains a problem here (non-linearity). Yet, instability has been acknowledged during training. Several attempts had to be done in order to get a more stable model by manually tuning a fixed initialization seed.

Neural networks are known to be good function approximators and, at first look, the functions they are approximating are likely to be nonlinear. In particular, an LSTM creates an embedding that transforms the function into a linear one for the final prediction. However this is not related to the fact that the input is nonlinear, which is the case for the data distribution of Covid-19. Actually, we conjecture that the input data obey a certain pattern, otherwise no model could approximate it. The limited latent space is also a problem, even more for the long sequences, as is the case here. Besides modeling well the long-term memories, it fails in regularizing for other sequences with different properties [22]. That is to say that, if a certain situation (lock-down or distancing) is kept, thus it could perform better. Besides, the problem of learning long-term dependencies remains as one of the main challenges in deep learning [22]. A last problem with LSTM is that the time series has to be stationary and, with stable mean, an assumption that does not hold with the data that we analyzed in this paper.

An issue that recalled attention is that for states approaching the peak, the final curve fitting process performed better with a curve visually closer to data, as is the case reported in Figure 18 and Figure 19. Notice in Figure 20, for example, this may indicate that the values close to the peak might have lower values than the ones that are predicted by MAE approach. This can be confirmed when the peak is reached. If this is the case, some adjustment can be done in our method in order to account for this property, which is our first idea for future works.

The clustering approach that is proposed in this paper uses a feature representation focusing on the early response of the countries. This was based on the assumption that the first week of the spread of the disease are crucial to determine its dynamics in a given region. However, in future work, it might be interesting refine the groups based on the most recent data, in order to obtain even more accurate predictions. For example, if a state is at six weeks after outbreak, we could compute the features for weeks four to six after outbreak.

We underline that, up to date, the predictions for Brazilian states obtained by our model (introduced in Section 3.4) tend to be closer than the predictions obtained from traditional methods, as for example using SEIR and SIR [19,34] as depicted in Section 3.1. In relation to the approaches SIRD and SIRASD [19], the first (SIRD) presented the peak at 17 June with about 40 millions of individuals predicted to be symptomatic infected at the pandemic end (according to interpretation of Figure 4 of their paper [19]). The second approach (SIRASD) predicted the peak at 10 June with about 10 million total symptomatic infected (still high valued). The peak date is about the currently expected (May–June); nonetheless, the numbers are very high in relation to the current ongoing numbers. In fact, they argue that for the short-term forecasts their results are in great accordance with data and conclude that the long-term forecasts may help them to discuss different types of social distancing policies.

Another work dealing with Brazilian data is briefly discussed now, where the parameters that were obtained for the unstructured SEIR model were used to adjust the parameters for an age-structured SEIR model. A multiplicative factor of 0.0125 for the number of deaths is considered in order to estimate the number of infected cases, since the work deals with relative proportions of death rates for the age-structured model. As a result, the total number of cases is predicted as being 18,959 for the Brazil Northeastern. The current number (as of 22 June) for this Brazilian region is 16,467, which is almost reaching that predicted number and it looks like it will get a much higher value, approximately double that by following current conditions.

Finally, based on another data-driven approach, we provide a comparison with the Bayesian Monte Carlo approach [35], also developed dealing with Brazilian regions. Besides aiming to having good performance in a time-series cross validation with respect to three days ahead forecasting (so short-term prediction), for a long-term estimation to the end of August the total numbers of contaminated seems to be overestimated in comparison to ours. Looking at the current values, with predictions starting as of 4 May 2020, our numbers are closer to the true numbers, to date.

## 5. Conclusions

This paper presents a data-driven model for predicting the Covid-19 dynamics, which uses cases that have occurred in other locations or countries with similar contamination patterns. Although our study focuses on the Brazilian reality, the proposed approach can be applied elsewhere, as long as there exists more advanced regions for learning. In order to determine which countries share similar distributions, we first applied a clustering algorithm to all available countries and sub-regions. This clustering was a key element in order to avoid training the predictive model for a given state on a country that has a totally different dynamics. In future work, we intend to refine these clusters periodically in order to account for the evolution of the strategies of the different regions studied. Subsequently, to predict the Covid-19 dynamics for the different Brazilian states, a data driven approach based on MAE was used. Our results show that this approach performs better than traditional methods and LSTM networks.

Hence, according to our studies, the possibility of using information from other regions (including other countries or states) with a more advanced pandemic stage in order to predict data for a region that is at a less advanced contagion stage, as far as they have somewhat near dynamic features was verified. We notice, for example, that some regions in Italy have had similar behaviors as in Brazil, for example Milan and São Paulo (and New York). We are aware that the approach is not perfect and that it has uncertainty errors; nonetheless, it indicates a good research direction.

Thus, based on the results discussed above, at the time of writing this paper, we could verify the applicability of data driven approaches to model Covid-19 dynamics. With this approach, city managers can get more precise information and better insight to plan their actions. Complementary material for this work can be found at www.natalnet.br/covid, where the next step is to implement this approach running and updating automatically, while using the most recent available data.

## Figures and Tables

**Figure 1 ijerph-17-05115-f001:**
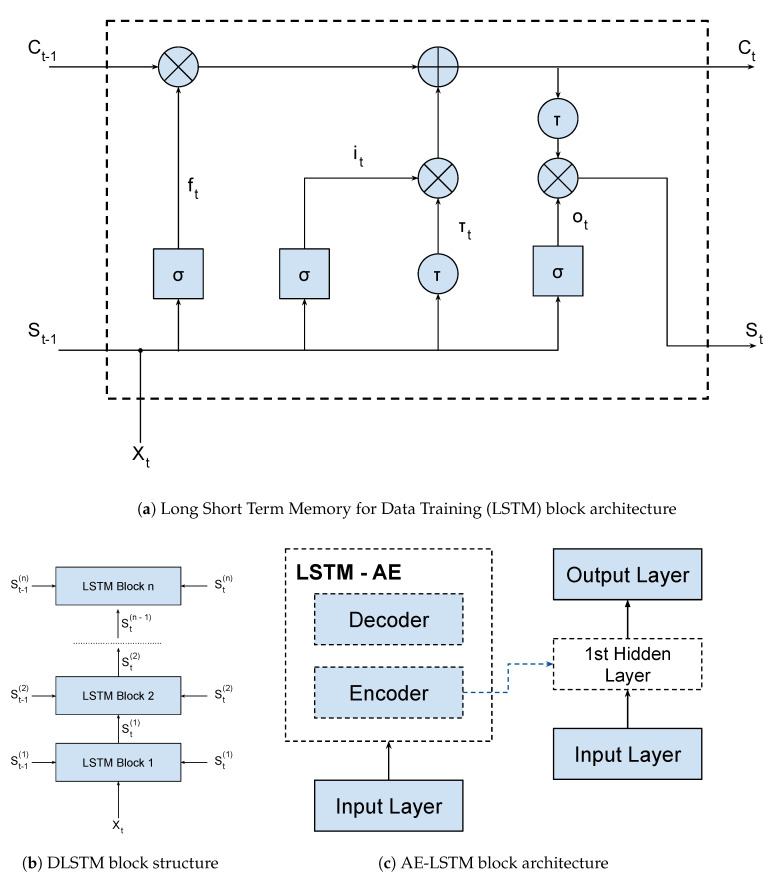
LSTM, DLSTM, and LSTM-SAE Blocks.

**Figure 2 ijerph-17-05115-f002:**
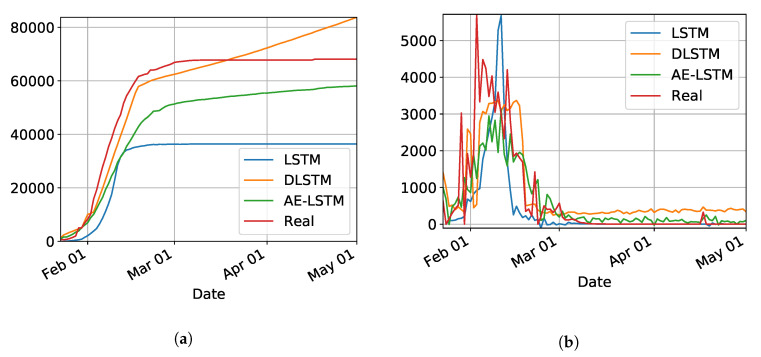
Comparison results for LSTM, DLSTM, and LSTM-SAE on Covid-19 cumulative (**a**) and daily (**b**) number of cases, data from Hubei, province of China.

**Figure 3 ijerph-17-05115-f003:**
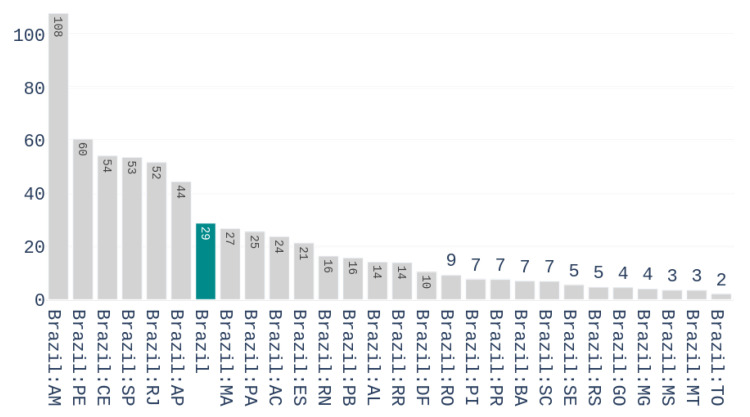
Number of deaths per million inhabitants in the different Brazilian states on 1 May 2020.

**Figure 4 ijerph-17-05115-f004:**
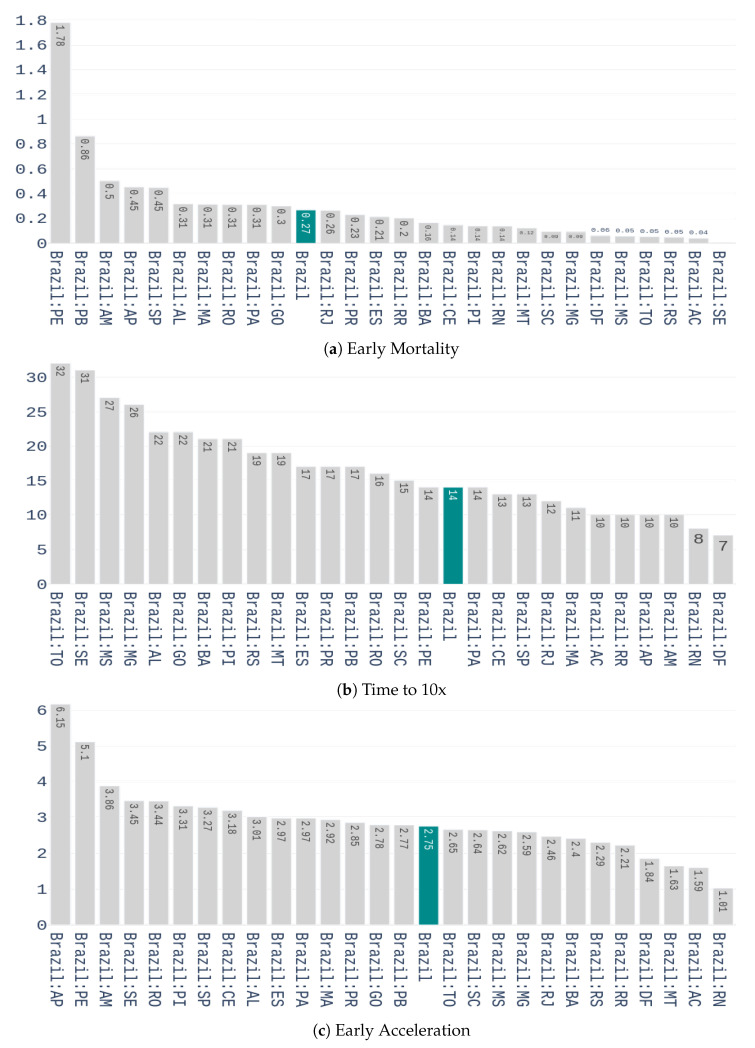
Values of the three features used for characterizing the early response to covid-19 for the Brazilian states.

**Figure 5 ijerph-17-05115-f005:**
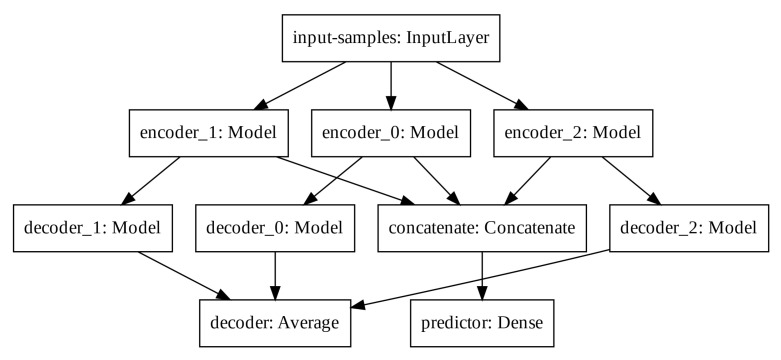
Modified Auto-Encoder architecture.

**Figure 6 ijerph-17-05115-f006:**
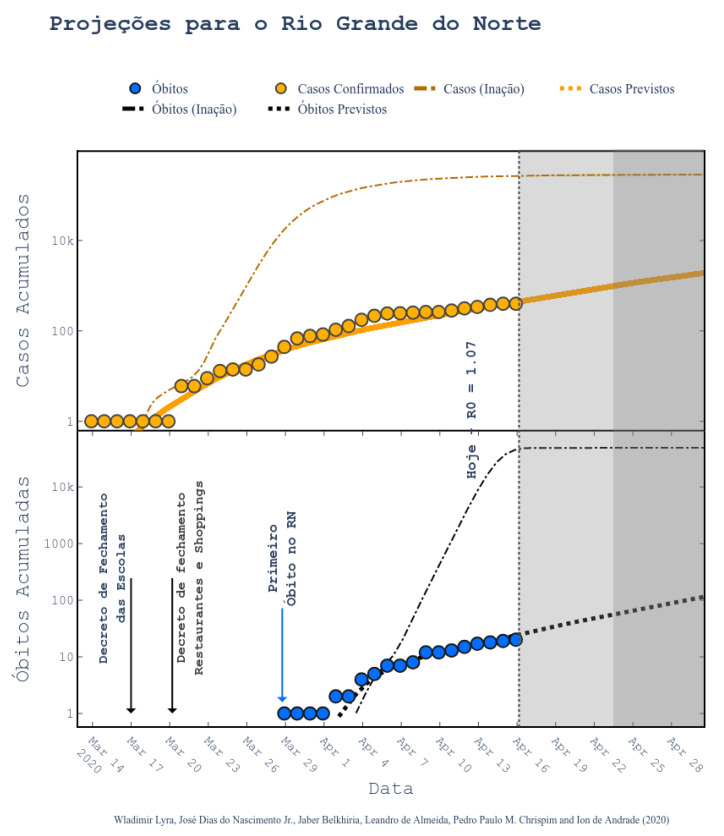
Projections for Rio Grande do Norte state (at the northeast of Brazil) [34]. Figure printed out from the web application running at http://astro.dfte.ufrn.br/html/Cliente/COVID19.php. Acessed on 4 May.

**Figure 7 ijerph-17-05115-f007:**
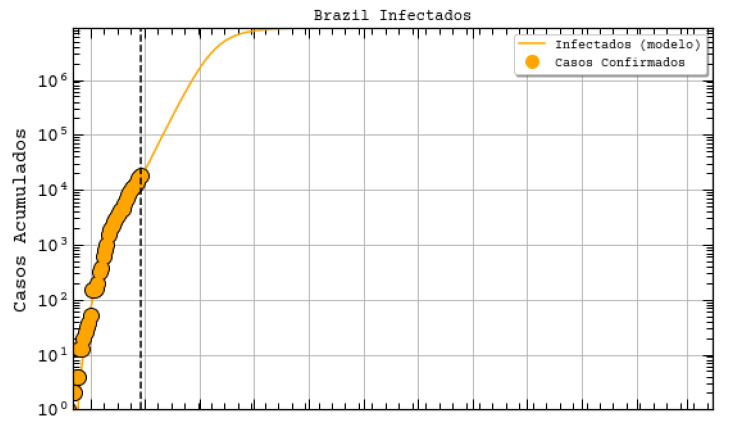
Projections for Brazil with adapted SEIR model [34], extracted from http://astro.dfte.ufrn.br/html/Cliente/COVID19.php. Acessed on 4 May.

**Figure 8 ijerph-17-05115-f008:**
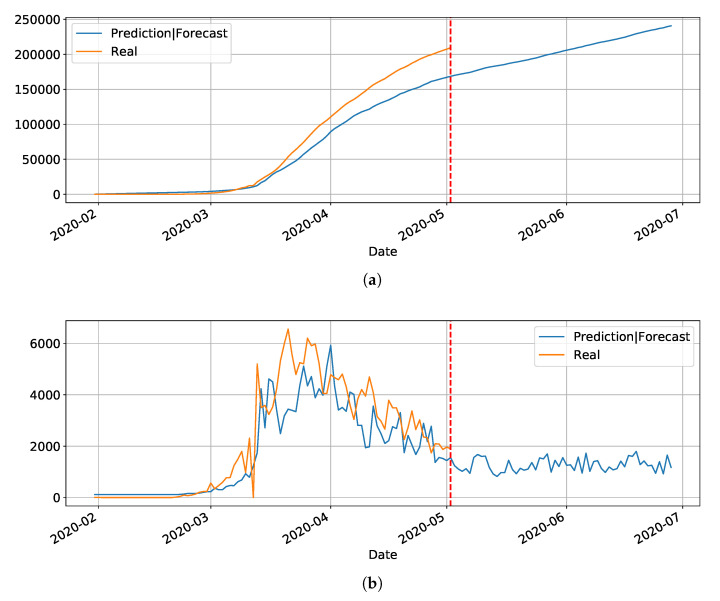
Predictions and forecasting to Italy on Covid-19 cumulative (**a**) and daily (**b**).

**Figure 9 ijerph-17-05115-f009:**
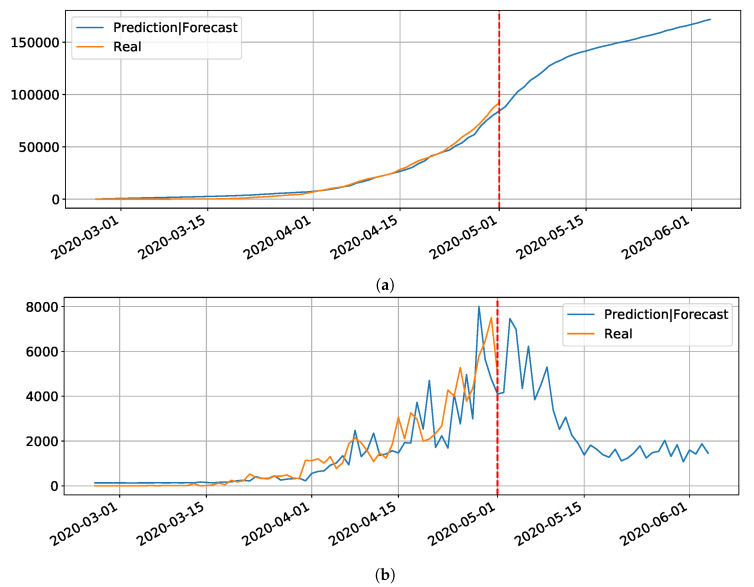
Predictions and forecasting to Brazil on Covid-19 cumulative (**a**) and daily (**b**).

**Figure 10 ijerph-17-05115-f010:**
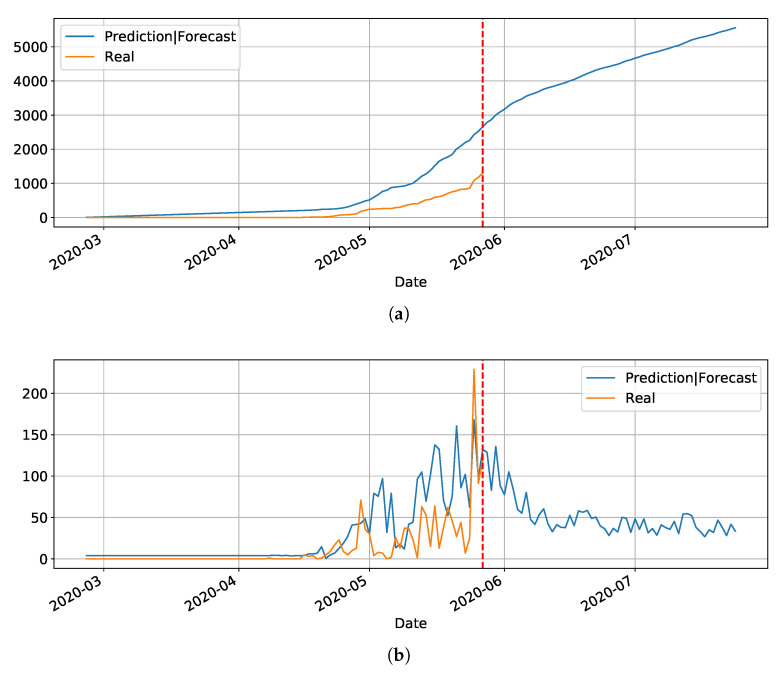
Predictions and forecasting to RN on Covid-19 cumulative (**a**) and daily (**b**).

**Figure 11 ijerph-17-05115-f011:**
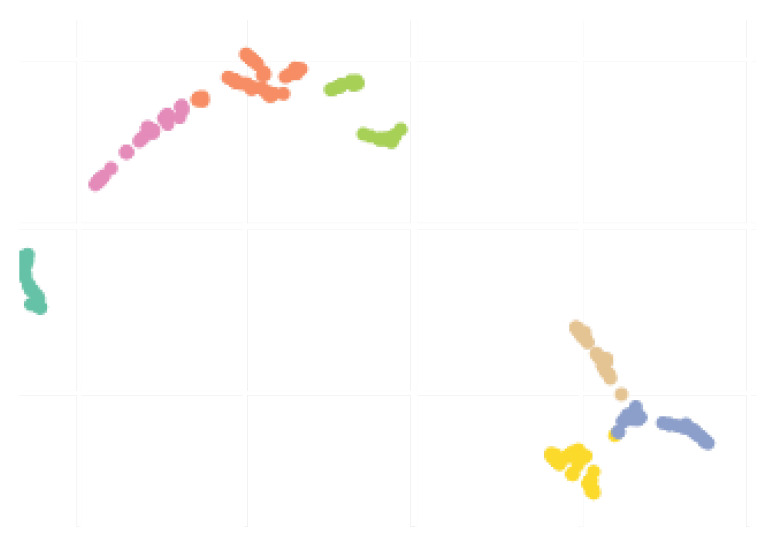
2D UMAP embedding of the different countries and states studied. The colors represents different clusters generated using Affinity Propagation.

**Figure 12 ijerph-17-05115-f012:**
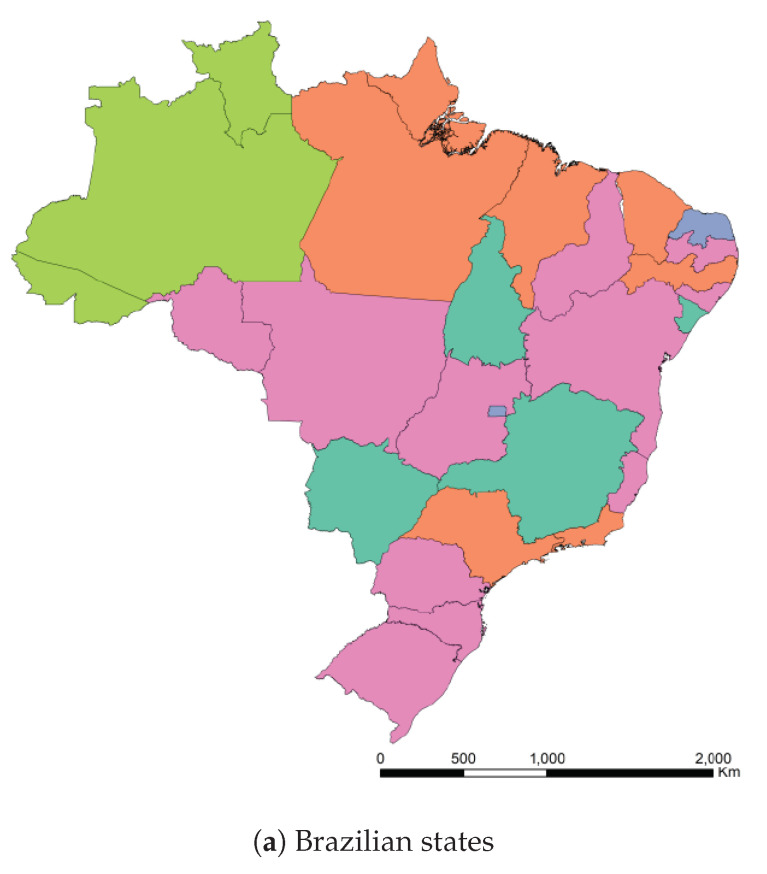
Clusters assignment of the different Brazilian states and world countries.

**Figure 13 ijerph-17-05115-f013:**
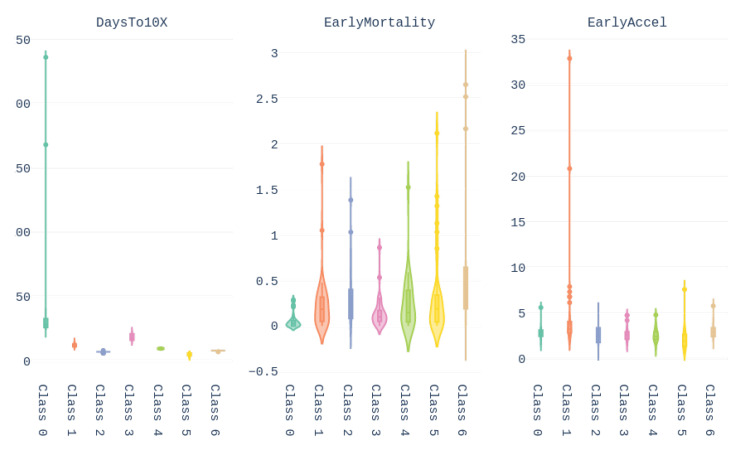
Violin plots representing the values taken by the different features for each groups obtained after UMAP + Affinity Propagation clustering.

**Figure 14 ijerph-17-05115-f014:**
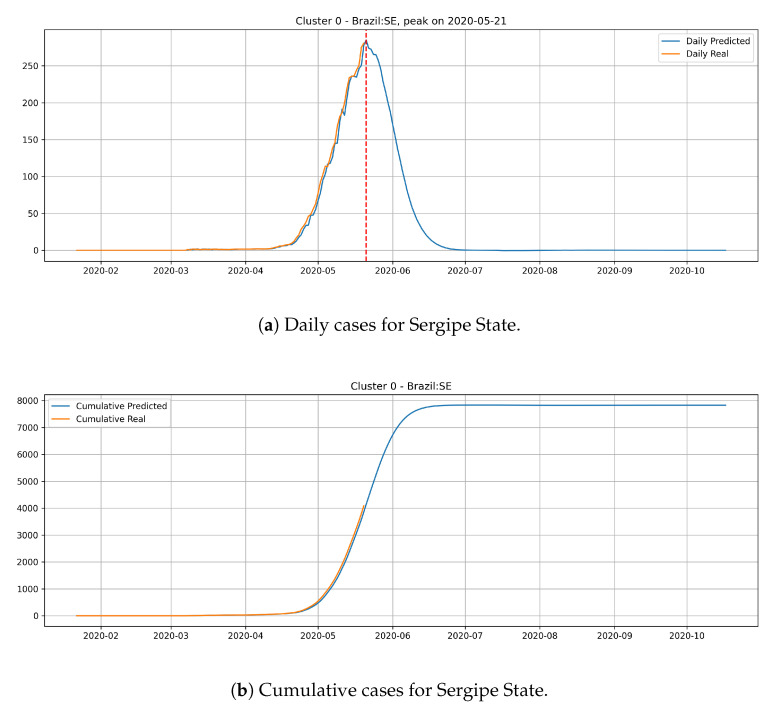
Daily and Cumulative cases for Sergipe State from the Cluster 0.

**Figure 15 ijerph-17-05115-f015:**
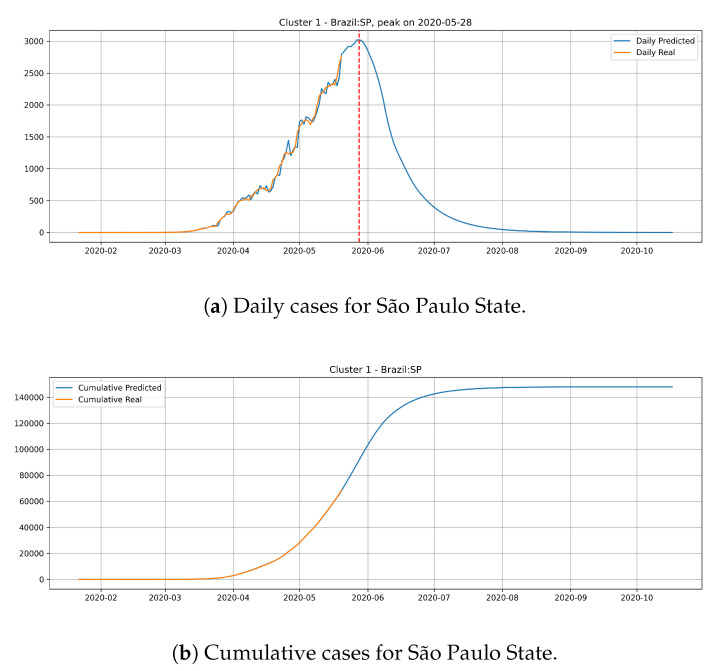
Daily and Cumulative cases for São Paulo State from the Cluster 1.

**Figure 16 ijerph-17-05115-f016:**
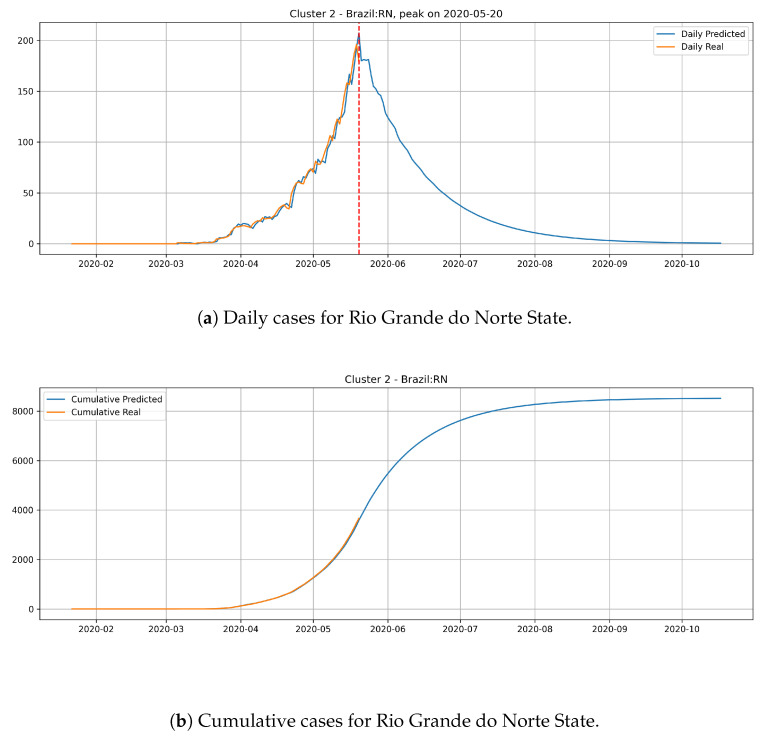
Daily and Cumulative cases for Rio Grande do Norte State from the Cluster 2.

**Figure 17 ijerph-17-05115-f017:**
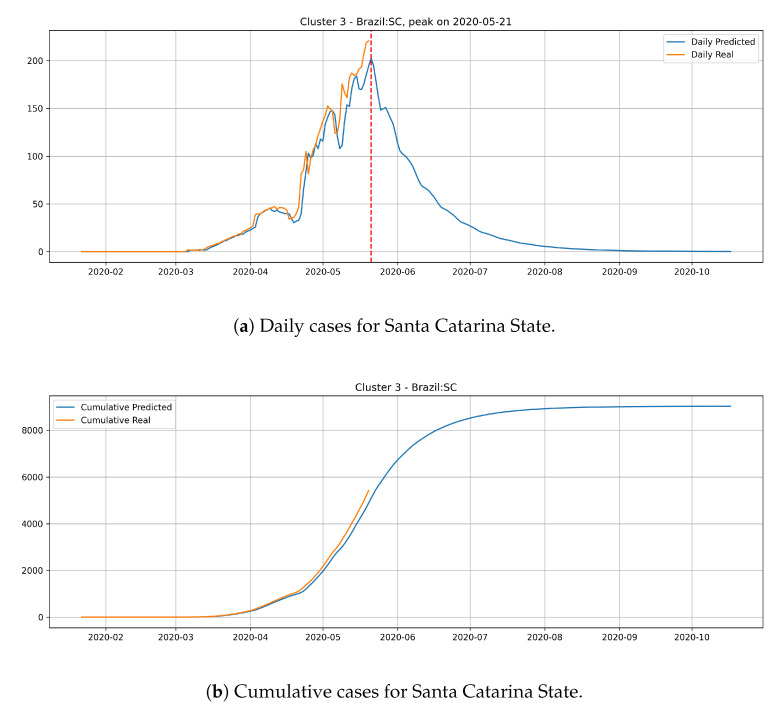
Daily and Cumulative cases for Santa Catarina State from the Cluster 3.

**Figure 18 ijerph-17-05115-f018:**
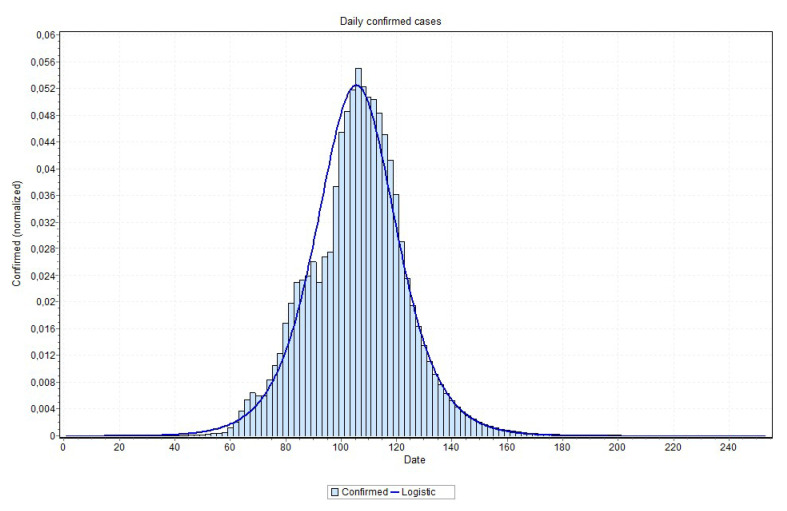
Curve fitting for Rio de Janeiro state (logNormal model was the best fit) with peak indicated on 31 May 2020.

**Figure 19 ijerph-17-05115-f019:**
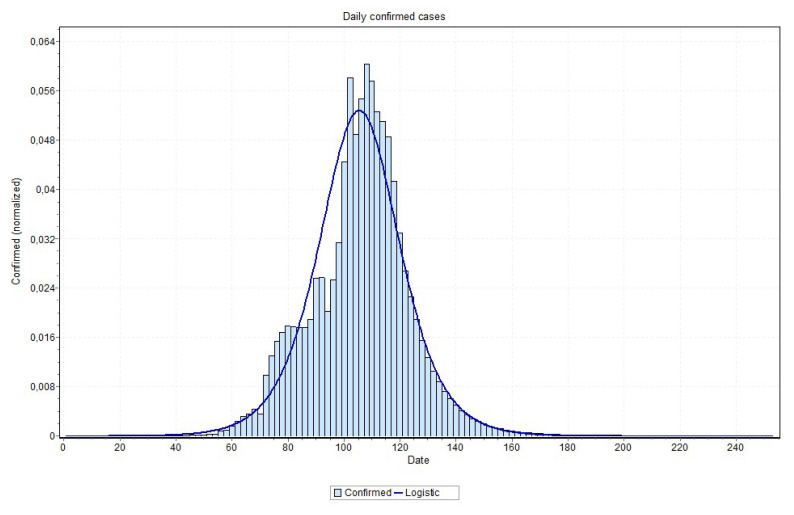
Curve fitting for São Paulo state (logistic model was the best fit) with peak indicated on 26 May 2020.

**Figure 20 ijerph-17-05115-f020:**
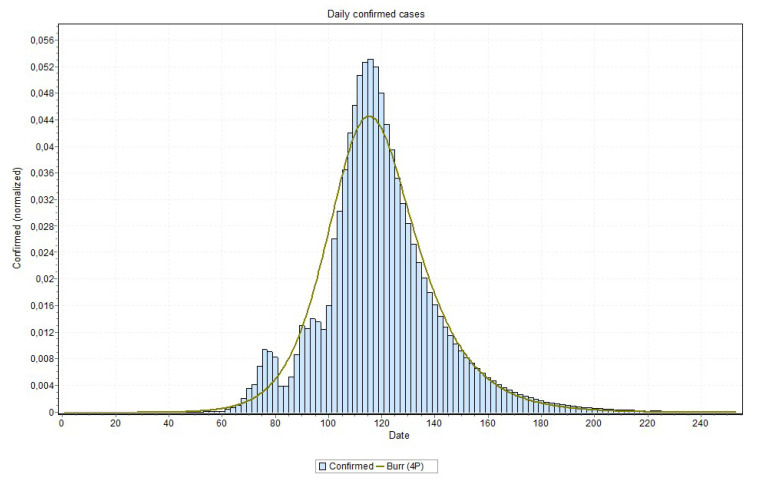
Curve fitting for Rio Grande do Norte state (Burr model was the best fit) with peak indicated on 21 May 2020.

**Table 1 ijerph-17-05115-t001:** Training parameters and Metrics.

	Parameters						Metrics	
	**Hidden Layers**	**Epochs**	**Epochs AE Model**	**Dropout**	**Units**	**Sequence Lenght**	**MAPE**	**Corelation**
**LSTM**	1	15	-	0.3	4	5	211	0.732
**DLSTM**	3	15	-	0.3	[10, 8, 6]	5	92	0.798
**LSTM-SAE**	1	50	15	0.3	4	5	84	0.822

**Table 2 ijerph-17-05115-t002:** Peak occurrences for each state predicted by the MAE Model and by a distribution probability. We also indicate the total number of cases expected by the MAE prediction and the day that it will reach 97% of the total number of cases.

State	Predicted by MAE	Curve Fit Peak	Best Curve	Total	97% of Total
TO	2020-05-20	2020-05-20	Pearson	846	2020-06-13
SE	2020-05-21	2020-05-22	Lognormal	2546	2020-06-13
MG	2020-05-21	2020-05-17	Logistic	2992	2020-06-03
MS	2020-05-21	2020-05-24	Pearson	327	2020-05-28
PA	2020-06-01	2020-06-03	Pearson	10,332	2020-06-11
AP	2020-05-20	2020-05-20	Logistic	5172	2020-06-15
MA	2020-05-14	2020-05-14	Logistic	9684	2020-06-10
CE	2020-05-30	2020-05-28	Pearson	11,556	2020-05-29
PE	2020-05-29	2020-05-30	Pearson	18,210	2020-06-08
RJ	2020-05-31	2020-05-31	Lognormal	21,587	2020-06-07
SP	2020-05-28	2020-05-26	Logistic	64,984	2020-06-07
RN	2020-05-20	2020-05-21	Burr	6025	2020-07-06
DF	2020-05-25	2020-05-27	Logistic	6347	2020-07-06
RO	2020-05-21	2020-05-24	Pearson	3061	2020-08-10
PI	2020-05-20	2020-05-24	Pearson	4974	2020-08-13
PB	2020-05-21	2020-05-26	Pearson	8765	2020-08-14
AL	2020-05-21	2020-05-28	Pearson	8119	2020-08-11
BA	2020-05-19	2020-05-19	Pearson	8945	2020-08-04
ES	2020-05-21	2020-05-23	Pearson	18,271	2020-08-12
PR	2020-05-21	2020-05-19	Lognormal	4038	2020-08-04
SC	2020-05-21	2020-05-26	Pearson	15,329	2020-08-13
RS	2020-05-20	2020-05-20	Gamma	4269	2020-08-03
MT	2020-05-21	2020-05-25	Pearson	701	2020-07-30
GO	2020-05-22	2020-05-25	Pearson	2245	2020-08-03
AC	2020-05-20	2020-05-25	Burr	3490	2020-07-12
AM	2020-05-20	2020-05-26	Burr	16,053	2020-07-04
RR	2020-05-20	2020-05-24	Burr	2206	2020-07-07

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
