# Peer review of "Forecasting Covid-19 Dynamics in Brazil: A Data Driven Approach"

_ijerph, 2020, doi:10.3390/ijerph17145115_

Round 1

Reviewer 1 Report

The presentation is unclear. What does the sigma function mean? 

Reviewer 2 Report

This work is devoted to develop a method to predict the dynamics of transmission of viral epidemics by analyzing contamination data from the perspective of artificial intelligence. They designed a modified auto-encoder network that is then trained from initial clustering of the world regions and learns to predict future data for Brazilian states.The research is interesting and has certain merits. However, the paper needs a little improvement before acceptance for publication. My detailed comments are as follows: (1) More clear and detail description shall be given especially the motivation and the significance in contributions of this research. This may be given in the beginning of the paper. (2) Comparison among similar works shall be given. Authors shall describe how significance this research it is by giving a comprehensive comparison with traditional approaches. (3) The English of your manuscript must be improved. I strongly suggest that you obtain assistance from a colleague who is well-versed in English or whose native language is English.

Reviewer 3 Report

In the paper the pandemic dynamics is investigated from the perspective of artificial intelligence. Nevertheless the authors are trying to analyze other approaches in Section 2.1. Their conclusions and comparisons with traditional methods could be much more complete after taken into account some recent investigations of the pandemic dynamics with the use of SIR model:

1. Statistics-based predictions of coronavirus epidemic spreading in mainland China. Innov Biosyst Bioeng. 2020; 4(1):13–18. DOI: 10.20535/ibb.2020.4.1.195074.

  1. Comparison of the coronavirus epidemic dynamics in Italy and mainland China [Preprint.] MEDRXIV. 2020 March. DOI: https://medrxiv.org/cgi/content/short/2020.03.18.20038133v

Do the results presented in Fig. 2 take into account the “jump” in the number of cases in China which happened on February 12 (see [1])? Have the fact that Chinese data before February 10 was not complete could influence the conclusions?

The epidemic conditions are changing permanently (for example, quarantines relax). Is it taken into account in the study? As far is I know the quarantine in Brazil differs from Chinese or Italian. Is it possible to use the information from other countries in order to make conclusions for Brazil? 

Round 2

Reviewer 1 Report

The second version of the paper is much improved version of the original one. I belive the paper is already fit to be published. 

Reviewer 3 Report

I am satisfied with the answers my questions and the corrections of the text. Please make the reference [13] by adding the name of journal and number:

13. Nesteruk I. Statistics-based predictions of coronavirus epidemic spreading in mainland China. Innov Biosyst Bioeng. 2020; 4(1):13–18. DOI: 10.20535/ibb.2020.4.1.195074.